# MRT-Lattice Boltzmann Model for Multilayer Shallow Water Flow

**Kevin R. Tubbs [1] and Frank T.-C. Tsai [2],***  

[1]  Engineering Science Program, Louisiana State University, 2228 Patrick F. Taylor Hall, Baton Rouge, LA 70803, USA

[2]  Department of Civil and Environmental Engineering, Louisiana State University, 3325 Patrick F. Taylor Hall, Baton Rouge, LA 70803, USA

*  Correspondence: ftsai@lsu.edu; Tel.: +1-225-578-4246; Fax: +1-225-578-4945

**Abstract:** The objectives of this study are to introduce a multiple-relaxation-time (MRT) lattice Boltzmann model (LBM) to simulate multilayer shallow water flows and to introduce graphics processing unit (GPU) computing to accelerate the lattice Boltzmann model. Using multiple relaxation times in the lattice Boltzmann model has an advantage of handling very low kinematic viscosity without causing a stability problem in the shallow water equations. This study develops a multilayer MRT-LBM to solve the multilayer Saint-Venant equations to obtain horizontal flow velocities in various depths. In the multilayer MRT-LBM, vertical kinematic viscosity forcing is the key term to couple adjacent layers. We implemented the multilayer MRT-LBM to a GPU-based high-performance computing (HPC) architecture. The multilayer MRT-LBM was verified by analytical solutions for cases of wind-driven, density-driven, and combined circulations with non-uniform bathymetry. The results show good speedup and scalability for large problems. Numerical solutions compared well to the analytical solutions. The multilayer MRT-LBM is promising for simulating lateral and vertical distributions of the horizontal velocities in shallow water flow.

**Keywords:** lattice Boltzmann model; shallow water equations; GPU computing; multiple relaxation times

## 1. Introduction

The shallow water equations are used to describe flow in bodies of water where the horizontal length scales are much greater than the fluid depth (e.g., river or lake hydrodynamics, coastal and estuarine circulation, overland flow, etc.). They have wide applications in hydraulic engineering [1–3] and coastal engineering [4], and can be used to study main physical phenomena of interest to scientists and engineers such as storm surges [5], tsunami and bore wave propagation [6], the stationary hydraulic jump, forces acting on off-shore structures, and river, reservoir and open channel flows [1,7]. The shallow water equations can also be coupled to transport equations to model pollutant transport [8], salinity and temperature [9], and sediment transport [6,10], which are important subjects in many industrial and environmental projects. More advanced depth-averaged models were developed for shear shallow water flows [11,12] and for coastal waves in the shoaling and surf zones [13,14].

When vertical effects are important, such as in baroclinic regimes, where density varies with salinity and temperature, three-dimensional flow should be used. This would require the solution of a system of equations coupling the Navier–Stokes equation to a moving free surface boundary. However, solving full Navier–Stokes equations in three dimensions is computationally challenging and may have difficulties handling the discontinuities in the free surface. Substantial literature exists on the application of various numerical methods, e.g., finite difference methods (FDMs), finite

volume methods (FVMs), and finite element methods (FEMs) to the three-dimensional shallow water equations [15–17].

The multilayer shallow water equations under the hydrostatic assumption present an alternative solution to the free surface Navier–Stokes system and lead to a precise description of the vertical profile of the horizontal velocity while preserving the robustness and computational efficiency of the shallow water equations. This study introduces an emerging mesoscopic numerical method, the lattice Boltzmann model, to simulate multilayer shallow water flows.

## 1.1. Lattice Boltzmann Model

Modeling of problems in hydrodynamics, hydraulics, and environmental fluid mechanics may be undertaken at three different length scales, commonly referred to as the microscopic, mesoscopic, and macroscopic levels [18]. Mesoscopic models based on the Boltzmann equation can be categorized into two sub-classes: continuous Boltzmann models and discrete Boltzmann models such as the lattice Boltzmann model. The main difference is that the distribution function is either continuous or discrete in particle velocity.

The lattice Boltzmann model can be seen as an alternative numerical scheme for simulating a wide range of fluid dynamics. The lattice Boltzmann model (LBM) was originally created to model flows governed by the Navier–Stokes equations [19–23]. This is because LBM is simpler to formulate, apply, and implement, including in high-performance computing (HPC) environments, than Riemann solvers because they do not require characteristic decomposition.

More recently, the Boltzmann-based methods have developed into an alternative and promising numerical technique for a wide range of computational fluid dynamics (CFD) techniques [24], including shock waves in compressible flows [25], multicomponent and multiphase flows in complex geometries [19,24], transcritical flows [26] and turbulent flows [27]. The LBM to porous medium flow includes Richard's unsaturated flow equation [28,29] and flow and mass transport [30–33]. The LBM has found a wide range of applications in a variety of fields involving shallow water equations. The LBM has been successfully adopted to study shallow water equations of wind-driven ocean circulation [34], intrusion of salt wedges in estuarine zones [35], continuous change in bed elevation [36], wetting–drying problems [37], two immiscible shallow layers of different density [38], and multilayer shallow water equations [39].

## 1.2. LBM on HPC Environments

The advantages of the LBM are its ease in parallelization because of the locality of particle interaction and the transport of particle information. The implementation of LBM on central processing unit (CPU)-based systems has been researched and numerous improvements are possible starting from standard LBM implementations [40,41]. The implementation of LBM on CPU-based architectures is achieved on both distributed and shared memory systems. LUDWIG [42] is a parallel LBM code for fluids, implementing message passing interface (MPI) to achieve full portability and good efficiency on both massively parallel processors (MPP) and symmetric multiprocessing (SMP) systems. With OpenMP [43], the LBM has been optimized to be implemented on multiple CPUs with shared-memory parallel programming [44]. More recently, the LBM has been a good candidate for implementation on hardware-accelerated systems using Graphics Processing Units (GPUs). It has been accelerated on a single GPU [45,46] or a GPU cluster [47,48] with MPI using the Compute Unified Device Architecture (CUDA).

## 1.3. Objective of the Study

The objective of this study is to develop a multiple-relaxation time (MRT)-LBM to simulate multilayer shallow water equations under GPU high-performance computing. This attempt is essential to extend the full capability of LBM to shallow water flow studies. Tubbs and Tsai [46] showed that single-relaxation-time (SRT)-LBM has a serious stability problem for solving shallow water

equations with very small viscosity. This stability would be even worse in the multilayer shallow water equations. This study will introduce MTR-LBM to increase stability and accuracy and eliminate spurious oscillations. Moreover, this study aims to investigate LBM performance on GPU-based HPC environments using MATLAB code and the Jacket GPU engine.

## 2. Multilayer Shallow Water Equations

Consider a shallow water flow regime in which the vertical length scale is much smaller than the horizontal length scale. By depth-integrating the continuity equation and the Navier–Stokes equations for incompressible and viscous flows with free surface, the shallow water equations with horizontal viscous terms are [3]:

$$\frac{\partial h}{\partial t} + \frac{\partial (h u_i)}{\partial x_i} = 0 \tag{1}$$

$$\frac{\partial (h u_i)}{\partial t} + \frac{\partial \left( h u_i u_j \right)}{\partial x_j} + \frac{\partial}{\partial x_i} \left( \frac{1}{2} g h^2 \right) = \nu \left[ \frac{\partial^2 (h u_i)}{\partial x_j \partial x_j} \right] + F_i \tag{2}$$

where $i$ and $j$ are Cartesian indices and the Einstein summation convention is used, $h$ is the water depth, $u_i$ is the depth-averaged velocity component in the $i$ direction, $g$ is the gravitational acceleration, $\nu$ is the kinematic viscosity, $F_i$ is the external force acting on the shallow water flow, and $t$ is the time. Based on the multilayer Saint-Venant system [49,50], the governing equations are similar to the above shallow water equations with horizontal viscous terms for each layer and additional terms for transferring momentum between layers:

$$\frac{\partial h^{(\ell)}}{\partial t} + \frac{\partial \left( h^{(\ell)} u_i^{(\ell)} \right)}{\partial x_i} = 0 \tag{3}$$

$$\frac{\partial \left( h^{(\ell)} u_i^{(\ell)} \right)}{\partial t} + \frac{\partial \left( h^{(\ell)} u_i^{(\ell)} u_j^{(\ell)} \right)}{\partial x_j} + \frac{\partial}{\partial x_i} \left( \frac{1}{2} g h^{(\ell)} H \right) = \nu \left[ \frac{\partial^2 \left( h^{(\ell)} u_i^{(\ell)} \right)}{\partial x_j \partial x_j} \right] + F_i^{(\ell)} \quad , \ell = 1, 2, \ldots, M \tag{4}$$

where $h^{(\ell)}$ is the local water height in layer $\ell$, $u_i^{(\ell)}$ is the local velocity component in the $i$ direction in layer $\ell$, $F_i^{(\ell)}$ is the external force acting on layer $\ell$ $H = \sum_{m=1}^{M_L} h^{(m)}$ is the water depth, and $M_L$ is the number of layers.

Six external forces are included in the model. They are wind-driven forcing ($F_{Wi}^{(\ell)}$) at the top-most layer, bed slope forcing ($F_{Pi}^{(\ell)}$), vertical kinematic eddy viscosity forcing ($F_{\mu i}^{(\ell)}$), non-conservative pressure source ($F_{NCi}^{(\ell)}$) [49–51], density gradient forcing, $\left( F_\rho^{(\ell)} \right)$ [52], and forcing from the Coriolis effect ($F_{Ci}^{(\ell)}$) as follows:

$$F_i^{(\ell)} = F_{Wi}^{(\ell)} + F_{Pi}^{(\ell)} + F_{\mu i}^{(\ell)} + F_{NCi}^{(\ell)} + F_{\rho i}^{(\ell)} + F_{Ci}^{(\ell)} \tag{5}$$

$$F_{Wi}^{(\ell)} = \delta_{M\ell} \frac{\tau_i^W}{\rho} = \delta_{M\ell} \frac{\rho_a C_W U_{Wi} W_s}{\rho} \tag{6}$$

where $\delta_{M\ell}$ is the Kronecker delta function ($\delta_{M\ell} = 1$, if $\ell = M$) and $\tau_i^W$ is the wind stress in $i$ direction, which is the product of fluid density ($\rho$), air density ($\rho_a$), wind stress coefficient ($C_W$), wind speed measured at 10 m above water surface ($W_s$), and wind velocity component in $i$ direction ($U_{Wi}$).

$$F_{Pi}^{(\ell)} = -g h^{(\ell)} \frac{\partial z_b}{\partial x_i} \tag{7}$$

where $z_b$ is the bed elevation.

$$F_{\mu i}^{(\ell)} = -\kappa \delta_{1\ell} u_i^{(\ell)} + 2\mu(1 - \delta_{M\ell})\frac{u_i^{(\ell+1)} - u_i^{(\ell)}}{h^{(\ell+1)} + h^{(\ell)}} - 2\mu(1 - \delta_{1\ell})\frac{u_i^{(\ell)} - u_i^{(\ell-1)}}{h^{(\ell)} + h^{(\ell-1)}} \tag{8}$$

where $\delta_{1\ell}$ is Kronecker delta function ($\delta_{1\ell} = 1$, if $\ell = 1$), $\kappa$ is the bottom friction coefficient, and $\mu$ is the vertical (kinematic) eddy viscosity, which is also called vertical viscosity [49–51] or internal friction coefficient [52]. The linear bed friction law is chosen to calculate bed friction. The 2nd and 3rd terms on the right-hand side of Equation (8) represent internal friction between layers. The vertical eddy viscosity μ is much larger than the kinematic viscosity $\nu$ in Equation (4).

$$F_{NCi}^{(\ell)} = \frac{gH^2}{2}\frac{\partial}{\partial x_i}\left(\frac{h^{(\ell)}}{H}\right) \tag{9}$$

$$F_{\rho i}^{(\ell)} = \frac{gh^{(\ell)}}{\rho}\int_H^{\bar{z}}\frac{\partial\rho}{\partial x_i}dz \tag{10}$$

where $\bar{z} = \frac{1}{2}h^{(\ell)} + \sum_{m=1}^{\ell-1} h^{(m)}$ is at the center of layer $\ell$. The fluid density is constant in this study. However, this study follows [52] to include longitudinal density gradient in order to study the baroclinic circulation.

$$F_{Ci}^{(\ell)} = \begin{cases} f_c h^{(\ell)} u_y, & i = x \\ -f_c h^{(\ell)} u_x, & i = y \end{cases} \tag{11}$$

where $f_c = 2\bar{\omega}\sin\varphi$ is the Coriolis parameter, which a function of Earth rotation rate ($\bar{\omega}$) and latitude ($\varphi$).

## 3. MRT-Lattice Boltzmann Modeling

This study adopts the multiple-relaxation-time (MRT) lattice Boltzmann model [53,54] to solve the multilayer shallow water equations. Specifically, the authors implement the MRT-LBM to a D2Q9 lattice, which defines the streaming velocity as

$$c_\alpha = \begin{cases} (0,0) & \alpha = 0 \\ c\left[\cos\left(\frac{1}{4}(2\alpha - 2)\pi\right), \sin\left(\frac{1}{4}(2\alpha - 2)\pi\right)\right] & \alpha = 1,2,3,4 \\ \sqrt{2}c\left[\cos\left(\frac{1}{4}(2\alpha - 9)\pi\right), \sin\left(\frac{1}{4}(2\alpha - 9)\pi\right)\right] & \alpha = 5,6,7,8 \end{cases} \tag{12}$$

where $c_\alpha = \{c_{\alpha i}\}$ is a streaming velocity along a streaming direction defined by an index $\alpha$, and $c$ is the lattice speed. Given square lattice size $\Delta x$ and time step $\Delta t$, the lattice speed is $c = \Delta x/\Delta t$. Then, the evolution equation for the MRT-LBM on a D2Q9 lattice for each layer $\ell$ is

$$f^{(\ell)}(x_i + c_{\alpha i}\Delta t, t + \Delta t) - f^{(\ell)}(x_i, t) = -M^{-1}S\left[m^{(\ell)}(x_i, t) - m^{(\ell)eq}(x_i, t)\right] + \frac{\Delta t}{6c^2}F_\alpha^{(\ell)}(x_i, t) \tag{13}$$

where $f^{(\ell)} = \left\{f_\alpha^{(\ell)}, \alpha = 0,1,2,\ldots,8\right\} \in R^9$ is a vector of particle distribution functions for layer $\ell$ $m^{(\ell)} \in R^9$ and $m^{(\ell)eq} \in R^9$ are vectors of moments and their equilibria, respectively, for layer $\ell$, and $M \in R^{9\times9}$ is a transformation matrix that transforms the particle distribution functions and equilibrium distribution functions (EDFs) from velocity space to moment space, which makes $m^{(\ell)} = Mf^{(\ell)}$ and $m^{(\ell)eq} = Mf^{(\ell)eq}$. $f^{(\ell)eq} = \left\{f_\alpha^{(\ell)eq}, \alpha = 0,1,2,\ldots,8\right\} \in R^9$ is a vector of the EDFs for layer $\ell$. $S = diag(s_0, s_1, \ldots, s_8) \in R^{9\times9}$ is a diagonal matrix of multiple relaxation rates, where $s_\alpha$ are the relaxation rates. $F_\alpha^{(\ell)} = \left\{\sum_i c_{\alpha i}F_i^{(\ell)}, \alpha = 0,1,2\ldots,8\right\}$ is a vector of external forces along the $\alpha$ direction. The forcing terms $F_i^{(\ell)}$ are given by the central scheme [3].

The evolution Equation (13) consists of two steps: streaming and collision. At the left-hand side, particle transport is achieved by pure advection executed in the streaming velocity space. At the

right-hand side, particle collision is achieved by linear relaxation processes executed in the moment space. For each time step, particle distribution functions reach their neighboring nodes simultaneously through prescribed lattice connections. The transformation matrix $M$ for D2Q9 is given by Lallemand and Luo [22]:

$$
M = \begin{pmatrix} b_0^T \\ b_1^T \\ b_2^T \\ b_3^T \\ b_4^T \\ b_5^T \\ b_6^T \\ b_7^T \\ b_8^T \end{pmatrix} = \begin{pmatrix}
1 & 1 & 1 & 1 & 1 & 1 & 1 & 1 & 1 \\
-4 & -1 & -1 & -1 & -1 & 2 & 2 & 2 & 2 \\
4 & -2 & -2 & -2 & -2 & 1 & 1 & 1 & 1 \\
0 & 1 & 0 & -1 & 0 & 1 & -1 & -1 & 1 \\
0 & -2 & 0 & 2 & 0 & 1 & -1 & -1 & 1 \\
0 & 0 & 1 & 0 & -1 & 1 & 1 & -1 & -1 \\
0 & 0 & -2 & 0 & 2 & 1 & 1 & -1 & -1 \\
0 & 1 & -1 & 1 & -1 & 0 & 0 & 0 & 0 \\
0 & 0 & 0 & 0 & 0 & 1 & -1 & 1 & -1
\end{pmatrix} \tag{14}
$$

where the vectors $\{b_\alpha\}$ are mutually orthogonal [53,55]. Inserting matrices $M$ and $S$ into Equation (13), the evolution equation in the direction $\alpha$ for each layer becomes [54]

$$
f_\alpha^{(\ell)}(x_i + c_{\alpha i}\Delta t, t + \Delta t) = f_\alpha^{(\ell)}(x_i, t) - \sum_{\beta=0}^{8} \frac{s_\beta b_{\beta\alpha}}{\|b_\beta\|}\left[ m_\beta^{(\ell)}(x_i, t) - m_\beta^{(\ell)eq}(x_i, t) \right] + \frac{\Delta t}{6c^2}\sum_i c_{\alpha i}F_i^{(\ell)}(x_i, t) \tag{15}
$$

The moments $m^{(\ell)}$ applied to the shallow water equations for each layer are

$$
m^{(\ell)} = \left( h^{(\ell)}, e^{(\ell)}, \varepsilon^{(\ell)}, j_x^{(\ell)}, q_x^{(\ell)}, j_y^{(\ell)}, q_y^{(\ell)}, p_{xx}^{(\ell)}, p_{xy}^{(\ell)} \right) \tag{16}
$$

where $m_0^{(\ell)} = h^{(\ell)}$ is the water depth, $m_1^{(\ell)} = e^{(\ell)}$ is related to the total energy, $m_2^{(\ell)} = \varepsilon^{(\ell)}$ is related to the energy square, $\left( m_3^{(\ell)}, m_5^{(\ell)} \right) = \left( j_x^{(\ell)}, j_y^{(\ell)} \right) = \left( hu_x^{(\ell)}, hu_y^{(\ell)} \right)$ are the flow momenta, $\left( m_4^{(\ell)}, m_6^{(\ell)} \right) = \left( q_x^{(\ell)}, q_y^{(\ell)} \right)$ are related to the head flux, and $m_7^{(\ell)} = p_{xx}^{(\ell)}$ $m_8^{(\ell)} = p_{xy}^{(\ell)}$ are related to the diagonal and off-diagonal components of the stress tensor, respectively. When applied to the shallow water equations, the conserved moments are the water depth and the flow momenta in each layer:

$$
m_0^{(\ell)} = h^{(\ell)} = \sum_{\alpha=0}^{8} f_\alpha^{(\ell)} \tag{17}
$$

$$
m_3^{(\ell)} = h^{(\ell)}u_x^{(\ell)} = \sum_{\alpha=0}^{8} c_{\alpha x}f_\alpha^{(\ell)}, \quad m_5^{(\ell)} = h^{(\ell)}u_y^{(\ell)} = \sum_{\alpha=0}^{8} c_{\alpha y}f_\alpha^{(\ell)}. \tag{18}
$$

The remaining moments are not conserved quantities. The EDFs applied to the multilayer shallow water are Equation (17).

$$
f_\alpha^{(\ell)eq} = h^{(\ell)}\omega_\alpha \left( \frac{c_s^2}{c^2} + \frac{u_i^{(\ell)}c_{\alpha i}}{c^2} + \frac{3}{2}\frac{\left( u_i^{(\ell)}c_{\alpha i} \right)^2}{c^4} - \frac{1}{2}\frac{u_i^{(\ell)}u_i^{(\ell)}}{c^2} \right) , \quad \alpha > 0
$$
$$
f_0^{(\ell)eq} = h^{(\ell)} - \sum_{\alpha=1}^{8} f_\alpha^{(\ell)eq} \tag{19}
$$

where $c_s^2 = gH/2$ is known as the squared speed of sound and is obtained through the recovery of the shallow water equations up to second order by the Chapman–Enskog analysis. For the D2Q9 lattice, the weighting coefficient $\omega_\alpha = 1/3$ is for $\alpha = 1,2,3,4$ and $\omega_\alpha = 1/12$ is for $\alpha = 5,6,7,8$. Since each layer has

the same planar discretization, the weighting coefficients $\omega_\alpha$ remain the same for each layer. Using Equations (14) and (19), the equilibrium moments, $m^{(\ell)eq} = M f^{(\ell)eq}$, are [46]

$$m_1^{(\ell)eq} = -4h^{(\ell)} + \frac{3gh^{(\ell)2}}{c^2} + \frac{3h^{(\ell)}\left(u_x^{(\ell)2} + u_y^{(\ell)2}\right)}{c^2}, \quad m_2^{(\ell)eq} = 4h^{(\ell)} - \frac{9gh^{(\ell)2}}{2c^2} - \frac{3h^{(\ell)}\left(u_x^{(\ell)2} + u_y^{(\ell)2}\right)}{c^2} \quad (20)$$

$$m_4^{(\ell)eq} = -\frac{h^{(\ell)}u_x^{(\ell)}}{c}, \quad m_6^{(\ell)eq} = -\frac{h^{(\ell)}u_y^{(\ell)}}{c} \quad (21)$$

$$m_7^{(\ell)eq} = \frac{3h^{(\ell)}\left(u_x^{(\ell)2} - u_y^{(\ell)2}\right)}{c^2}, \quad m_8^{(\ell)eq} = \frac{h^{(\ell)}u_x^{(\ell)}u_y^{(\ell)}}{c^2} \quad (22)$$

The equilibria of the conserved moments ($m_0^{(\ell)}$, $m_3^{(\ell)}$, and $m_5^{(\ell)}$) are equal to themselves. Therefore, the relaxation rates $s_0$, $s_3$, and $s_5$ have no effect on MRT-LBM solutions. With the moment equilibria given by Equations (20)–(22), the shallow water equations can be recovered with the shear viscosity ($\nu$) and bulk viscosity ($\zeta$) given [55] as

$$\nu = \frac{1}{3}\left(\frac{1}{s_7} - \frac{1}{2}\right)c\Delta x = \frac{1}{3}\left(\frac{1}{s_8} - \frac{1}{2}\right)c\Delta x \quad (23)$$

$$\zeta = \frac{1}{6}\left(\frac{1}{s_1} - \frac{1}{2}\right)c\Delta x \quad (24)$$

Setting $s_\alpha = 1/\tau$, the evolution Equation (13) reduces to the SRT-LBM model, where $\tau$ is the relaxation time. The LBM with a single relaxation time is commonly referred to as the Bhatnagar–Gross–Krook (BGK) collision operator model (LBGK), and has been introduced to the shallow water equations [34,56]. For the SRT-LBM, it has $\nu = 2\zeta = \frac{1}{3}\left(\tau - \frac{1}{2}\right)c\Delta x$. When $\tau$ is close to 1/2, the kinematic viscosity becomes very small (e.g., $\nu = 1 \times 10^{-6}$ m$^2$/s), causing the numerical instability. The MRT-LBM model has no such problem because the relaxation rates can be selected to attain stable solutions. In addition, to increase solution stability, this study adopts an implicit step to update flow velocities [39].

## 4. Multilayer Initial and Boundary Conditions

### 4.1. Initial Conditions

The initial conditions for a physical problem to be modeled are given in the form of macroscopic variables, which is normal practice in traditional numerical methods. Since the lattice Boltzmann formulation is based on solving Equation (15), the initial conditions must be written in terms of the distribution function. Given the initial macroscopic boundary conditions, $h^{(\ell)}(x_i, t = 0)$, $u_x^{(\ell)}(x_i, t = 0)$, and $u_y^{(\ell)}(x_i, t = 0)$, the EDFs, $f_\alpha^{(\ell)eq}$ (Equation (19)), are computed and used as initial conditions for $f_\alpha^{(\ell)}$.

### 4.2. Periodic Boundary Conditions

In cases where the flow pattern repeats itself at boundaries, a periodic boundary condition is required. To achieve boundary conditions in the x direction in the lattice Boltzmann formulation, we set the unknown distribution functions, $f_1^{(\ell)}$, $f_5^{(\ell)}$ and $f_8^{(\ell)}$ at the inflow boundary, $\Gamma_{inflow}$ (see Figure 1a for an example) to be the same as known distribution functions at the outflow boundary, $\Gamma_{outflow}$:

$$f_\alpha^{(\ell)}\left(x_i \in \Gamma_{inflow}, t\right) = f_\alpha^{(\ell)}\left(x_i \in \Gamma_{outflow}, t\right), \quad \alpha = 1, 5, 8 \quad (25)$$

The unknown distribution functions, $f_3$, $f_6$ and $f_7$ at the outflow boundary are set to the corresponding known distribution functions at the inflow boundary:

$$f_\alpha^{(\ell)}\left(x_i \in \Gamma_{outflow}, t\right) = f_\alpha^{(\ell)}\left(x_i \in \Gamma_{inflow}, t\right), \quad \alpha = 3, 6, 7 \tag{26}$$

Periodic boundary conditions similar to Equations (25) and (26) in the y direction can be formulated.

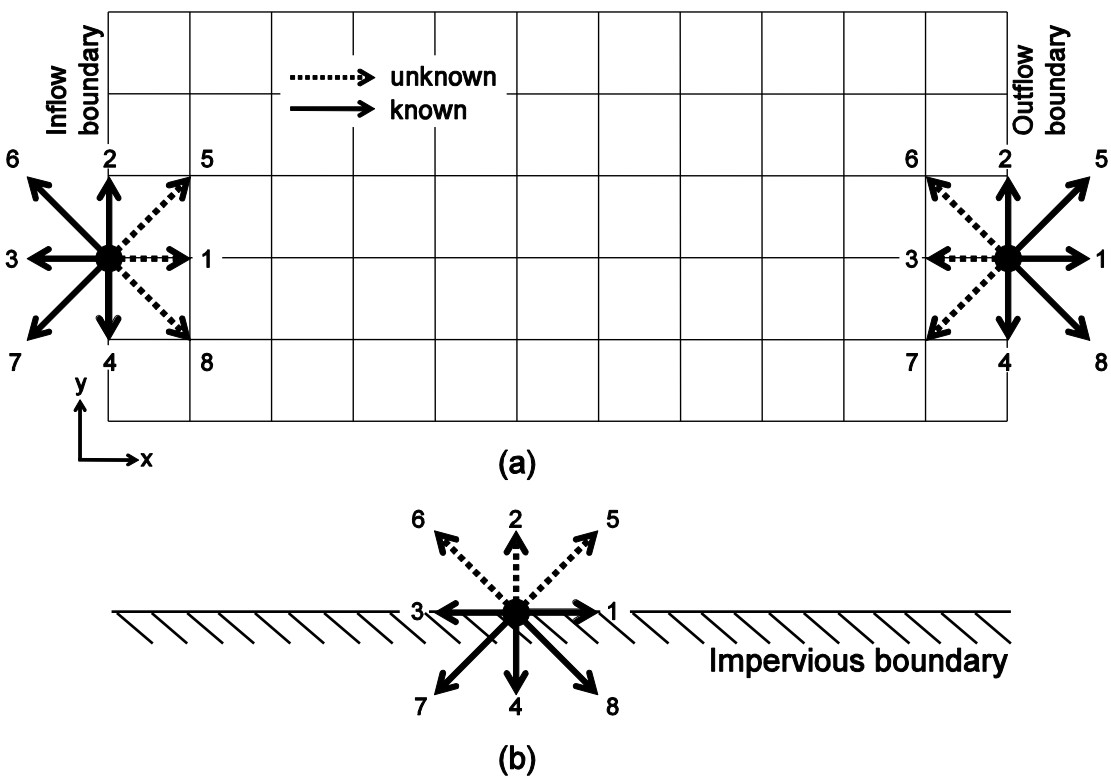

**Figure 1.** Lattice nodes for (**a**) periodic boundary or open boundary, and (**b**) impervious boundary.

### 4.3. Solid Boundary Conditions

Solid boundaries in the flow region can be either no-slip or free-slip, which corresponds to zero velocity or zero normal velocity, respectively, at the boundary. A no-slip boundary condition is achieved by the bounce-back scheme under symmetry conditions [21]. We set the unknown distribution functions, $f_2^{(\ell)}$, $f_5^{(\ell)}$ and $f_6^{(\ell)}$ at the solid boundary (see Figure 1b for example) to the known distributions, $f_4^{(\ell)}$, $f_7^{(\ell)}$ and $f_8^{(\ell)}$ corresponding to the opposite directions:

$$f_2^{(\ell)} = f_4^{(\ell)}, \quad f_5^{(\ell)} = f_7^{(\ell)}, \quad f_6^{(\ell)} = f_8^{(\ell)}. \tag{27}$$

This results in zero normal and tangential flow velocities at the boundary.

A free-slip boundary condition is also achieved by the bounce-back scheme [21]. The unknown distribution functions, $f_2^{(\ell)}$, $f_5^{(\ell)}$ and $f_6^{(\ell)}$ at the solid boundary is assigned to the known distributions, $f_4^{(\ell)}$, $f_8^{(\ell)}$, and $f_7^{(\ell)}$ corresponding to the reflected directions to the solid boundary:

$$f_2^{(\ell)} = f_4^{(\ell)}, \quad f_5^{(\ell)} = f_8^{(\ell)}, \quad f_6^{(\ell)} = f_7^{(\ell)}. \tag{28}$$

This ensures no flow across the normal direction and flow along the tangential direction.

*4.4. Open Boundary Conditions*

Open boundary conditions refer to known macroscopic boundary values or functions at boundaries (e.g., known water depth at inflow or outflow boundaries). If flow velocity and water depth at the boundaries are known, unknown distribution functions can be computed using Equations (17) and (18) following the method described by Zou and He [57]. For example, at the inflow boundary (see Figure 1a), Equations (17) and (18) lead to the following three equations:

$$f_0^{(\ell)} + f_1^{(\ell)} + f_2^{(\ell)} + f_3^{(\ell)} + f_4^{(\ell)} + f_5^{(\ell)} + f_6^{(\ell)} + f_7^{(\ell)} + f_8^{(\ell)} = h^{(\ell)} \tag{29}$$

$$c\left(f_1^{(\ell)} + f_5^{(\ell)} + f_8^{(\ell)}\right) - c\left(f_3^{(\ell)} + f_6^{(\ell)} + f_7^{(\ell)}\right) = h^{(\ell)} u_x^{(\ell)} \tag{30}$$

$$c\left(f_2^{(\ell)} + f_5^{(\ell)} + f_6^{(\ell)}\right) - c\left(f_4^{(\ell)} + f_7^{(\ell)} + f_8^{(\ell)}\right) = h^{(\ell)} u_y^{(\ell)} \tag{31}$$

Given known flow velocities and water heights in each layer in Equations (29)–(31), three unknown distribution functions, $f_1^{(\ell)}$, $f_5^{(\ell)}$ and $f_8^{(\ell)}$ can be determined. The same approach is applicable to determine three unknown distribution functions, $f_3^{(\ell)}$, $f_6^{(\ell)}$ and $f_7^{(\ell)}$, at the outflow boundary.

## 5. GPU Accelerated LBM

This study implements the MRT-LBM code to a GPU architecture to solve the multilayer shallow water equations. The MRT-LBM code was written in MATLAB® (2008, Natick, MA, USA) and is parallelized with AccelerEyes Jacket GPU Engine and the NVIDIA® Compute Unified Device Architecture (CUDA) [58] on a single GPU workstation. In all simulations, the computations were performed in a single precision. In this study, the simulations are performed on a single workstation 3.0-GHz Intel® Core™2 Extreme quad core with an NVIDIA® Tesla™ C1060 Computing Processor. The NVIDIA® Tesla™ C1060 Computing Processor contains 240 stream processors running at 1.3 GHz, which has a peak performance of 933 GFLOPs. Readers refer to [46] for detail explanations of how the GPU accesses memory and processes data.

*5.1. Jacket's GPU Engine*

The Jacket GPU engine for MATLAB® is built on NVIDIA's CUDA technology. It enables a standard MATLAB code to run on the GPU and connects the user-friendliness of MATLAB directly to the speed and visual computing capability of the GPU [59]. MATLAB GPU computing with Jacket starts at the most basic level through the replacement of low-level MATLAB data structures which normally reside on the CPU with data structures that reside on the GPU. This replaces the lowest level of MATLAB's CPU-bound computation engine, moving the work MATLAB would normally do on the CPU to the GPU. Jacket Beta version 0.3-20080710 [59] on a 32-bit Windows XP with MATLAB R2007b is used. Jacket is run on the 2.0 beta version of the CUDA toolkit for Windows, which uses version 1.1 compute capabilities.

Jacket-enabled MATLAB scripts achieve speed improvements in the range of 2–10 times, and in some cases up to 100x improvements, over equivalent CPU versions. While Jacket accelerates MATLAB functions and computations at a lower level, the overall speedup of an algorithm depends on the nature of the algorithm. The LBM has a very straightforward implementation consisting of only local calculations (collisions) and nearest neighbor memory transfers (streaming), which makes it a great candidate to be implemented both on the GPU and in MATLAB.

*5.2. Optimizing MATLAB GPU Performance*

Implementing algorithms on the GPU using Jacket requires certain considerations to optimize performance. Both MATLAB and Jacket perform best on vectorized code. A vectorized code can make it easy to determine which parts of an algorithm are inherently serial or parallel. Both MATLAB

and Jacket take advantage of the inherent parallelism of the MATLAB scripting M-language which is extremely powerful when utilized wisely. A good understanding of the memory dependencies of an algorithm is necessary as CPUs are inherently serial computing devices and GPUs are inherently parallel computing devices. In a program, one can control which segments of code are run on each device through the casting operations. Each casting operation to and from the GPU pushes or pulls data back and forth from CPU memory to GPU memory. Excessive memory transfers should be avoided as they will reduce the performance of an application. The Jacket software minimizes these memory transfers automatically in normal operation. However, care must be taken in implementing an algorithm. Fortunately, the LBM can be completely vectorized and therefore all computations can be carried out on the GPU. Transfers to CPU memory are only necessary for outputting solutions at desired intervals. Currently, a transfer to CPU is necessary for MATLAB plotting routines; however, due to the nature of the GPU, plots can be created through OpenGL.

*5.3. Computational Aspects*

The basic code to be parallelized on the GPU using Jacket is written in MATLAB M-Language and follows the same traditional practice of explicitly separating the collision and streaming operations. The solution algorithm has not changed. However, in order to take advantage of the GPU and MATLAB, the codes must be vectorized. Due to vectorization, the solution procedure focuses on three main steps: the calculation of local macroscopic variables from distribution functions, the collision step and the streaming step. Two copies of the distribution functions are used to allow the code to be vectorized. The vectorized version of the code is straightforward. The Jacket GPU engine makes translating the code on the GPU as simple as casting the variables to the GPU. From there, all calculations are performed on the GPU. Since the LBM is inherently parallel, there is no need to cast variables back to the CPU until the end of the simulation or when variables are written to file.

## 6. Numerical Experiments

*6.1. GPU Performance*

The parallel performance on the GPU is investigated in this section. A rectangular lake, 170 km × 60 km, with a flat bottom was used to simulate wind-driven circulation. The model domain was discretized into four different grid cell sizes for comparison. The numbers of columns, rows, and layers are: $171 \times 61 \times 10$, $341 \times 121 \times 10$, $681 \times 241 \times 10$, and $1361 \times 481 \times 10$ with cell sizes of 1000 m, 500 m, 250 m, and 125 m, respectively, and 10 layers. Each layer has an initial local water height of 1 m, such that initial water depth is 10 m. The initial flow velocity is zero. We used $\tau = 0.501$ and $c = 20$ m/s with $\Delta t$ calculated as $\Delta x/c$ for LBM parameters. Other parameters are $f_c = 0$ s$^{-1}$, $\rho = 1025$ kg/m$^3$, $\rho_a = 1.2$ kg/m$^3$, $C_W = 0.0015$, $\mu = 0.01$ m$^2$/s, $\kappa$ 0.001 m/s, $U_{Wx} = 7.4536$ m/s and $U_{Wy} = 0$ (positive x wind direction). According to Equation (6), wind stress is $\tau_x^W = 0.1$ N/m$^2$ along the x direction.

The EDFs in Equation (19) with static water and 1-m initial local water height determines the initial condition for the distribution functions. We applied free-slip bounce-back boundary conditions to the vertical walls.

The parallel performance of the GPU is based on arithmetic intensity and data access patterns [54]; therefore, the parallel performance is investigated based on the scalability of the speedup with increasing problem size. The simulations were run for a simulation time of 30 h where steady state has been achieved. The average time per time step is investigated to make a fair comparison on computational effort.

The execution time per time step and speedup for the GPU over a single core of the CPU in MATLAB for the four cases are shown in Table 1. It demonstrates the importance of arithmetic intensity in LBM performance on the GPU. If the number of lattice nodes is sufficiently high, the computations outweigh the data transferring overhead, yielding a high arithmetic intensity. The multilayer LB

algorithm yields a speedup of approximately 2.2-fold on the smallest number of lattice nodes with the maximum speedup of approximately 22.0-fold on the largest number of lattice nodes.

**Table 1.** The grid size, execution time per time step and speedup for the graphics processing unit (GPU) over a single core of the central processing unit (CPU) in MATLAB.

| Grid Size | CPU | | Speedup |
|---|---|---|---|
| | **Execution Time per Time Step (sec)** | | |
| $171 \times 61 \times 10$ | 0.44 | 0.19 | 2.2 |
| $341 \times 121 \times 10$ | 3.04 | 0.30 | 10.1 |
| $681 \times 241 \times 10$ | 14.19 | 0.95 | 14.9 |
| $1361 \times 481 \times 10$ | 56.60 | 2.57 | 22.0 |

*6.2. Verification*

We first verify the multilayer MRT-LBM using the steady-state analytical solutions in [52] by neglecting the effects of inertial terms and the unsteady terms for cases of wind-driven, density-driven and combined wind-driven and density-driven circulations. The velocity in the x direction is

$$u_x = g \frac{\partial H}{\partial x} \left\{ \frac{z^2 - H_0^2}{2\mu} - \frac{H_0}{\kappa} \right\} - \frac{g}{\rho} \frac{\partial \rho}{\partial x} \left\{ \frac{z^3 + H_0^3}{6\mu} + \frac{H_0^2}{2\kappa} \right\} + \frac{\tau_x^W}{\rho} \left\{ \frac{z + H_0}{\mu} + \frac{1}{\kappa} \right\} \tag{32}$$

where $H_0$ is the still water depth. The free-surface slope term $g\partial H/\partial x$ is

$$g \frac{\partial H}{\partial x} = \frac{-\frac{g}{\rho} \frac{\partial \rho}{\partial x} \left\{ \frac{H_0^4}{8\mu} + \frac{H_0^3}{2\kappa} \right\} + \frac{\tau_x^W}{\rho} \left\{ \frac{H_0^2}{2\mu} + \frac{H_0}{\kappa} \right\}}{\left\{ \frac{H_0^3}{3\mu} + \frac{H_0^2}{\kappa} \right\}} \tag{33}$$

The model is a 3400 m × 1400 m rectangular lake with a flat bottom. The initial water depth is 65 m. Three numerical models with difference number of layers were developed for comparison. The model domain was discretized into 501 × 206 lattices in the planar direction with a cell size of 6.8 m and five, ten and twenty layers. The corresponding initial local water heights for each layer is 13, 6.5, and 3.25 m, respectively. Fluid density is considered to be constant in this study but can have a constant density gradient along the longitudinal direction when density-driven circulation is considered. The LBM parameters are $\Delta t = 0.17$ s and $c = 40$ m/s. To achieve a kinematic viscosity of $v = 1 \times 10^{-6}$ m²/s, the relaxation time parameter in the SRT-LBM is given by $\tau = \frac{1}{2} + 3v/c\Delta x$, i.e., $\tau = 0.5 + 1.1029 \times 10^{-8}$. For the MRT-LBM, the relaxation rates $s_4 = s_6 = s_7 = s_8 = 1/\tau$, and $s_1 = s_2 = s_7 - 0.6$ were used. The linear friction law was used for bed friction. The approach to initialize distribution functions and the boundary conditions to the four vertical walls are the same as in the previous numerical case.

The wind-driven circulation validation is performed for two different wind stress values, $\tau_{iz}^W = 0.03$ N/m² and $\tau_i^W = 0.3$ N/m². Wind direction is along the positive $x$ direction. The physical parameters for this case are $\partial\rho/\partial x = 0$, $\partial\rho/\partial y = 0$, $\rho = 1025$ kg/m³, $\rho_a = 1.2$ kg/m³, $C_W = 0.0015$, $\mu = 0.004$ m²/s, $\kappa = 0.005$ m/s. As shown in Figure 2, the multilayer MRT-LBM solutions from the 5-layer model to the 20-layer model compare well to the analytical solutions for $u_x$ profile at the location (x, y) = (1700 m, 700 m) (the center of the lake). Figure 2 indicates that the 10-layer model is sufficient to capture the velocity profile.

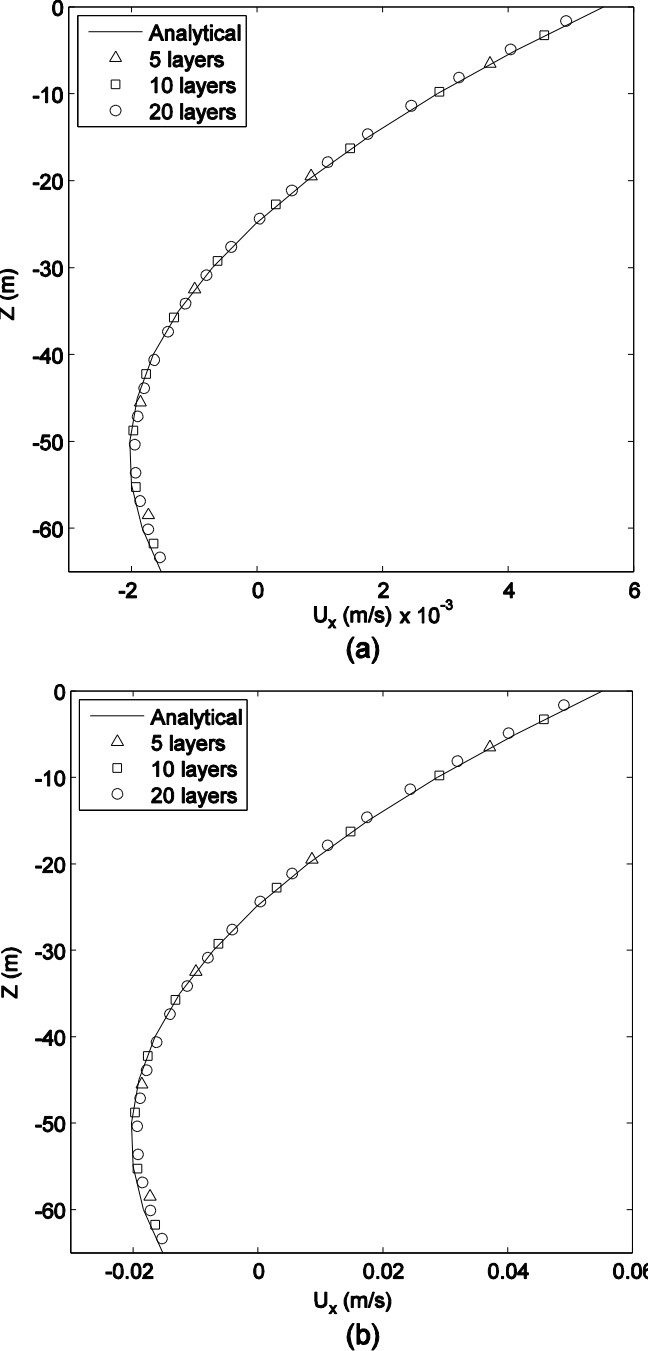

**Figure 2.** Comparisons of numerical model prediction with analytical solution for the wind-driven case: (**a**) $\tau_i^W = 0.03$ N/m$^2$, and (**b**) $\tau_i^W = 0.3$ N/m$^2$.

The density-driven circulation validation is performed for two different horizontal density gradients, $\partial\rho/\partial x = -5 \times 10^{-7}$ kg/m$^4$ and $\partial\rho/\partial x = -5\times10^{-5}$ kg/m$^4$. The physical parameters for this case are $\tau_i^W = 0$, $\partial\rho/\partial y = 0$, $\rho = 1025$ kg/m$^3$, $\rho_a = 1.2$ kg/m$^3$, $C_W = 0.0015$, $\mu = 0.004$ m$^2$/s, $\kappa = 0.005$ m/s. The multilayer MRT-LBM solutions with three different numbers of layers compare well to the analytical solutions for $u_x$ profiles for constant horizontal density as shown in Figure 3. Figure 3 indicates that the 10-layer model is sufficient to capture the velocity profile.

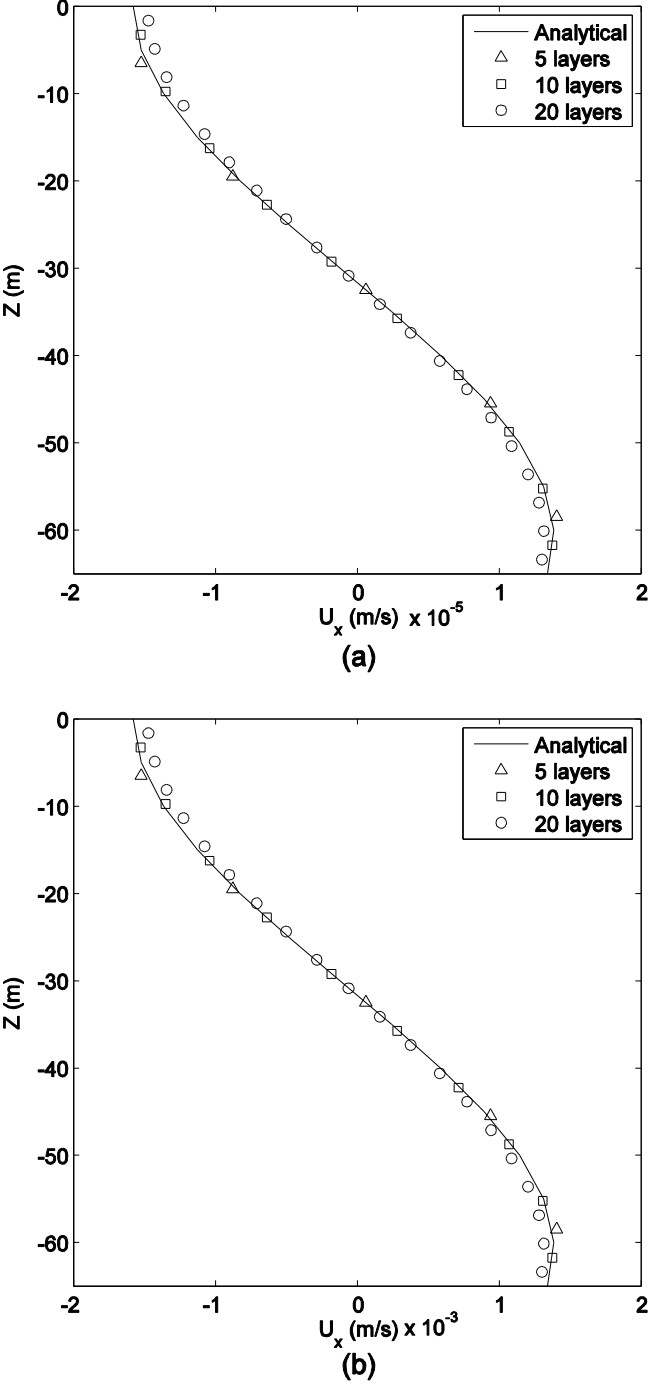

**Figure 3.** Comparisons of numerical model prediction with an analytical solution for the density-driven case: (**a**) $\partial\rho/\partial x = -5 \times 10^{-7}$ kg/m$^4$, and (**b**) $\partial\rho/\partial x = -5 \times 10^{-5}$ kg/m$^4$.

The multilayer MRT-LBM is also validated for combinations of wind-driven and density-driven circulation with $\tau_i^W = 0.03$ N/m$^2$, $\partial\rho/\partial x = -5 \times 10^{-5}$ kg/m$^4$ and $\tau_i^W = 0.3$ N/m$^2$, and $\partial\rho/\partial x = -5 \times 10^{-4}$ kg/m$^4$. The physical parameters for this case are $\partial\rho/\partial y = 0$, $\rho = 1025$ kg/m$^3$, $\rho_a = 1.2$ kg/m$^3$, $C_W = 0.0015$, $\mu = 0.004$ m$^2$/s, and $\kappa = 0.005$ m/s. With the positive $x$ wind direction, as shown in Figure 4, the multilayer MRT-LBM solutions compare well to the analytical solutions for $u_x$ profile for the combined effects. Figure 4 indicates that the 10-layer model is sufficient to capture the velocity profile.

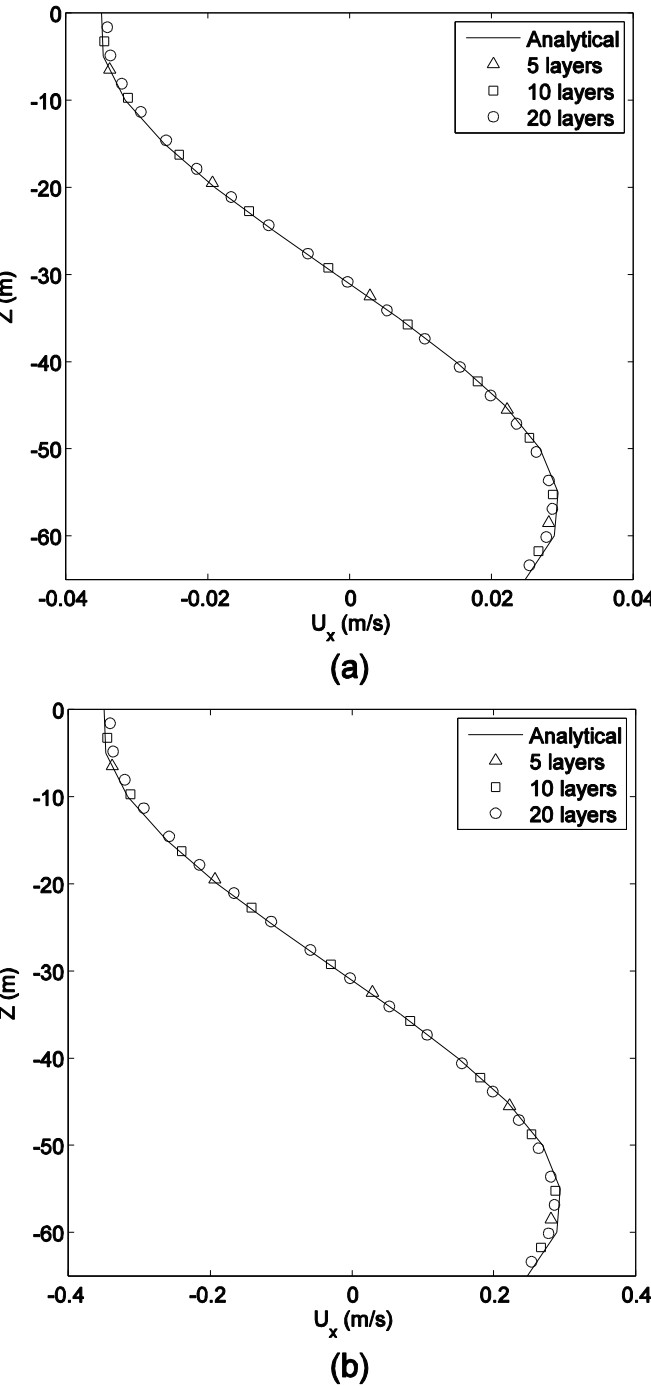

**Figure 4.** Comparisons of numerical model prediction with an analytical solution for the wind-density-driven case: (**a**) $\tau_i^W$ = 0.03 N/m², $\partial\rho/\partial x$ = −5×10⁻⁵ kg/m⁴, and (**b**) $\tau_i^W$ = 0.3 N/m², $\partial\rho/\partial x$ = −5×10⁻⁴ kg/m⁴.

*6.3. Wind-Driven and Density-Driven Circulation in Rotating Basins*

In this example, the multilayer MRT-LBM is demonstrated on GPU-based HPC. The study simulates wind-driven, density-driven, and combined wind-density-driven circulations over a Gaussian bathymetry profile [39]:

$$H_0(y) = 8 + 8exp\left[-\left(\frac{y - 3D/10}{2000}\right)^2\right] + 12exp\left[-\left(\frac{y + 3D/10}{2000}\right)^2\right] \tag{34}$$

where $D$ is the width of the basin. The $x$ axis coincides with the southern lateral wall of the basin and is pointed toward the head of the system. The $y$ axis is laid along the closed boundary at x = 0. The numerical domain is 100 by 15 km. The maximum depth, 20 m, occurs at y = −3D/10 km. The local maximum depth, 16 m, occurs at y = 3D/10 km.

For each case, the grid size is 801 × 121 × 10. The LBM parameters are $\Delta x$ = 125 m, $\Delta t$ = 6.25 s, and $c$ = 20 m/s. To achieve a kinematic viscosity of $\nu = 1 \times 10^{-6}$ m$^2$/s, the relaxation time parameter in the SRT-LBM is $\tau = 0.5 + 1.2 \times 10^{-9}$. For the MRT-LBM, the relaxation rates, $s_4 = s_6 = s_7 = s_8 = 1/\tau$, and $s_1 = s_2 = s_7 - 0.6$, were used. All closed boundaries have the free-slip bounce-back boundary condition. The initial water is stationary. The horizontal density gradient is constant. Uniform wind stress was linearly increased for the first six simulated hours and became constant after that. Wind direction is along positive $x$ direction. We executed numerical simulations on a single workstation with a 3.0-GHz Intel® Core™2 Extreme quad core processor and a NVIDIA® Tesla™ C1060 Computing Processor. The parallel performance of a single core of the quad core processor and the Tesla are compared.

For the wind-driven case, the physical parameters are $\tau_i^W$ = 0.04 N/m$^2$, $\partial\rho/\partial x = 0$, $\partial\rho/\partial y = 0$, $\rho$ = 1025 kg/m$^3$, $\rho_a$ = 1.2 kg/m$^3$, $C_W$ = 0.0015, $\mu$ = 0.004 m$^2$/s, $\kappa$ = 0.0025 m/s, and $f_c = 1\times10^{-4}$ s$^{-1}$. The wind velocity is $U_{Wx}$ = 4.7140 m/s and $U_{Wy}$ = 0. Figure 5a,b show the u$_x$ distributions at plane $x$ = 50 km and plane x = 98 km. The $u_x$ is in the same direction as wind at all shallow depths along the transverse boundaries and the center of the channel. However, $u_x$ is in the opposite direction of wind at deep depths. Figure 5c,d show the contours of the transverse flow $u_y$ at plane $x$ = 50 km and plant $x$ = 98 km. The Coriolis effect produces surface elevation contours with strong lateral variability.

For the density-driven case, the physical parameters are $\tau_i^W = 0$, $\partial\rho/\partial x = -5 \times 10^{-8}$ kg/m$^4$, $\partial\rho/\partial y = 0$, $\rho$ = 1025 kg/m$^3$, $\rho_a$ = 1.2 kg/m$^3$, $C_W$ = 0.0015, $\mu$ = 0.004 m$^2$/s, $\kappa$ = 0.005 m/s, and $f_c = 1 \times 10^{-4}$ s$^{-1}$. Figure 6 shows the $u_x$ and $u_y$ distributions at plane $x$ = 50 km and plane $x$ = 98 km. The $u_x$ flows in the direction of the horizontal gradient at all depths in the shallow regions along the transverse boundaries and the center of the channel shown in Figure 6a,b. The $u_x$ is in the opposite direction of horizontal gradient at deep depths. These flow directions are the opposite of those found in the purely wind-driven case. Highest flow occurs near the surface and flow decreases with depth due to the bottom friction. Figure 6c,d show the transverse flow $u_y$ at plane $x$ = 50 km and plane $x$ = 98 km. Although the flow field is reversed for this case, the effect of the Earth's rotation is consistent with the wind-driven case. Moreover, the transverse velocities exhibit similar behavior as in the wind-driven case with stronger magnitude at the $x$ = 98 km plane. However, at the $x$ = 50 km plane, the velocities are small, yet exhibit a vertical distribution of positive and negative flows.

For the combined wind-driven and density-driven case, the physical parameters are $\tau_i^W = 0.04$ N/m$^2$, $\partial\rho/\partial x = -5 \times 10^{-8}$ kg/m$^4$, $\partial\rho/\partial y = 0$, $\rho$ = 1025 kg/m$^3$, $\rho_a$ = 1.2 kg/m$^3$, $C_W$ = 0.0015, $\mu$ = 0.004 m$^2$/s, $\kappa$ = 0.005 m/s, and $f_c = 1 \times 10^{-4}$ s$^{-1}$. Figure 7 shows the $u_x$ and $u_y$ distributions at plane $x$ = 50 km and plane $x$ = 98 km. For the combined wind-driven and density-driven case, the $u_x$ distribution is similar to the density-driven case in terms of direction of the flow in shallow and deep regions as shown in Figure 7a,b. Figure 7c,d show the contours of the transverse flow, $u_y$ at $x$ = 50 km and $x$ = 98 km planes. The flow patterns created by bottom friction, the Earth's rotation, and bathymetry are all consistent with previous results. The main difference in the combined case is that the magnitudes of the flow are smallest near bed, increase in the positive $z$ direction, and then decrease again near the surface. This is due to the bottom friction and the wind stress occurring in the opposite direction of the density gradient. The density gradient accounts for the direction of the flow, while the wind stress accounts for the smaller magnitude velocities near the surface.

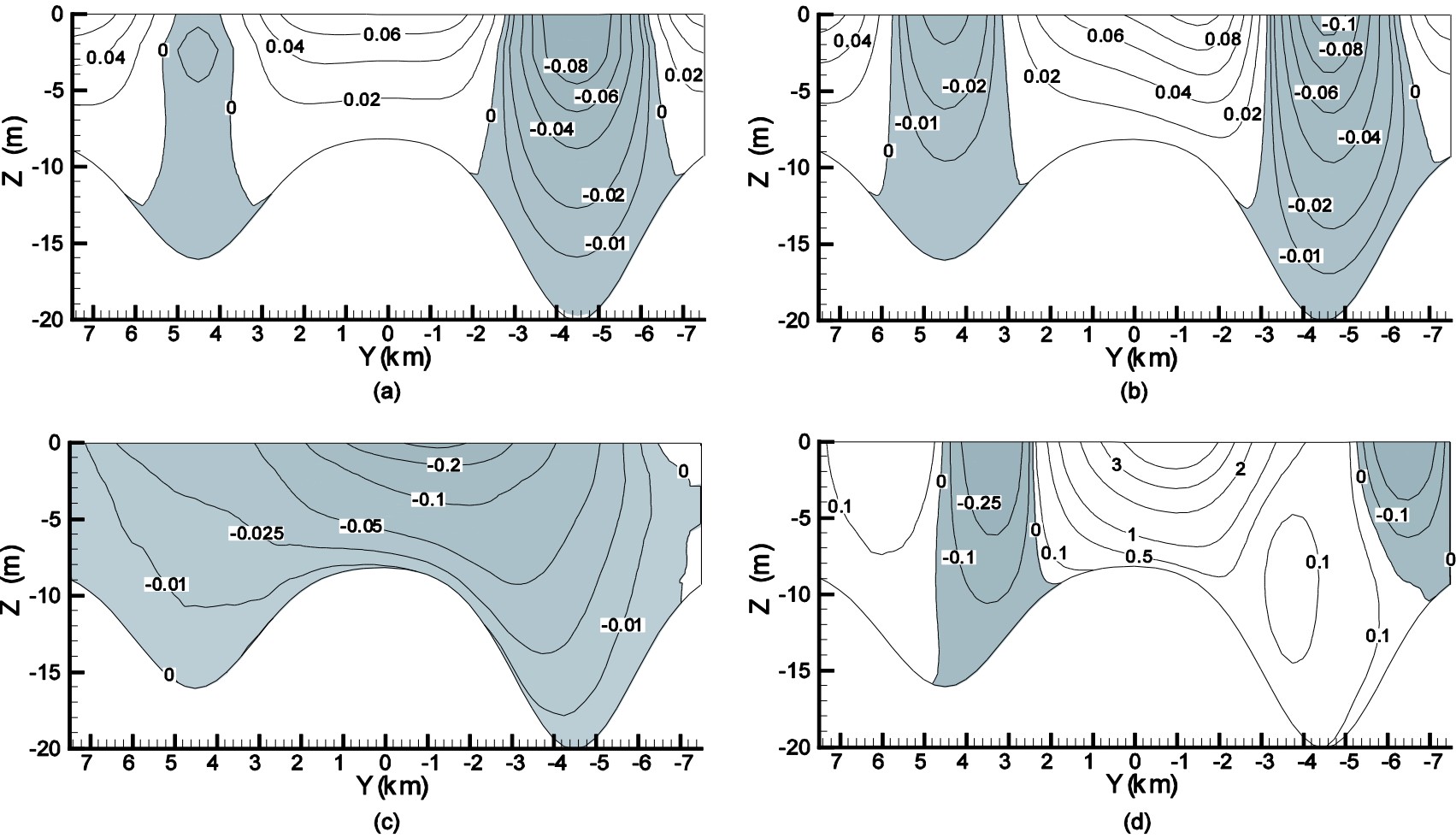

**Figure 5.** Contours of $u_x$ velocity (m/s) at (**a**) $x = 50$ km and (**b**) $x = 98$ km and contours of $u_y$ velocity (m/s $\times 10^{-2}$) at (**c**) $x = 50$ km and (**d**) $x = 98$ km for the wind-driven case. The dark areas represent negative velocities

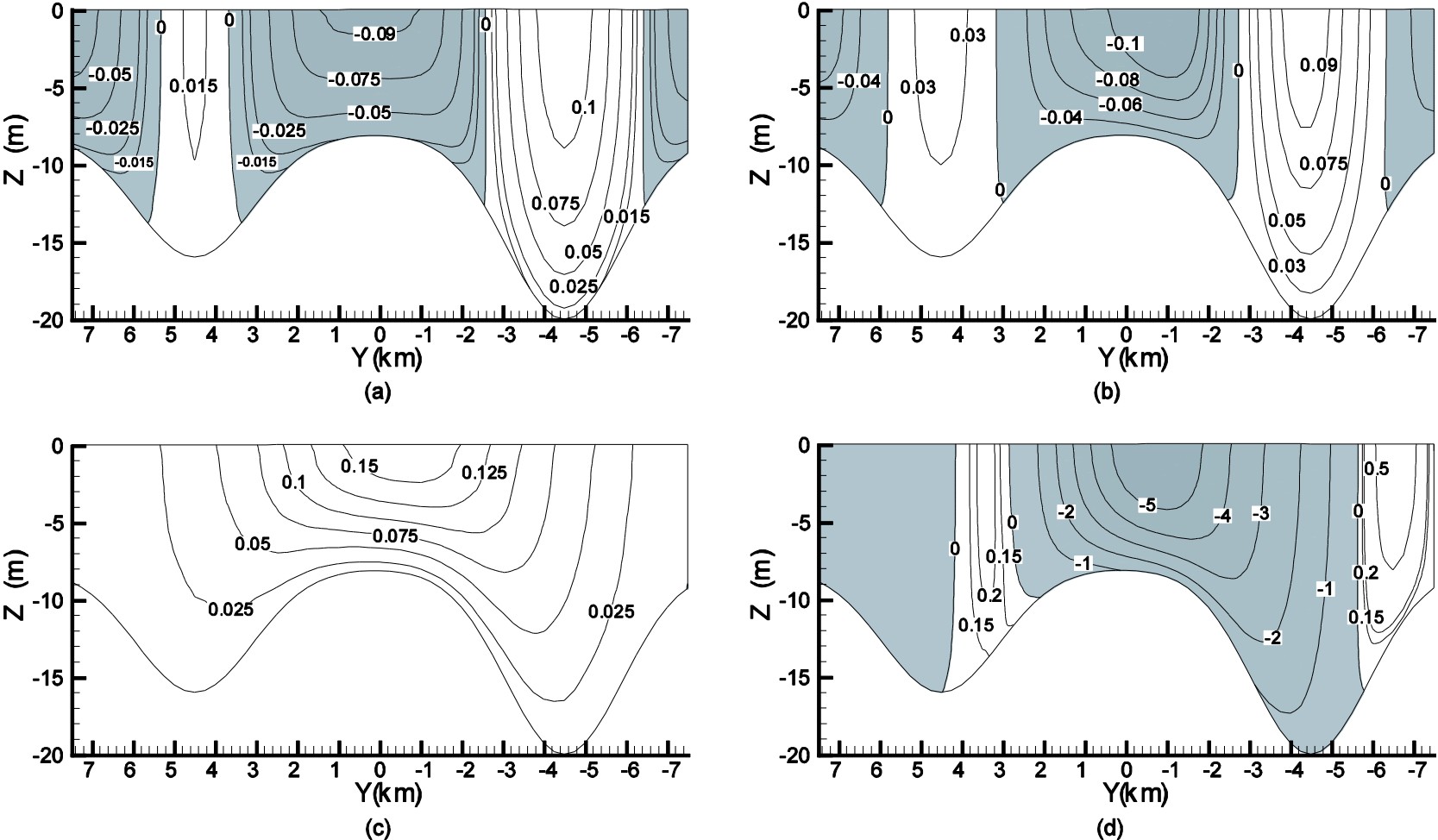

**Figure 6.** Contours of $u_x$ velocity (m/s) at (**a**) $x = 50$ km and (**b**) $x = 98$ km and contours of $u_y$ velocity (m/s $\times 10^{-2}$) at (**c**) $x = 50$ km and (**d**) x = 98 km for the density-driven case. The dark areas represent negative velocities.

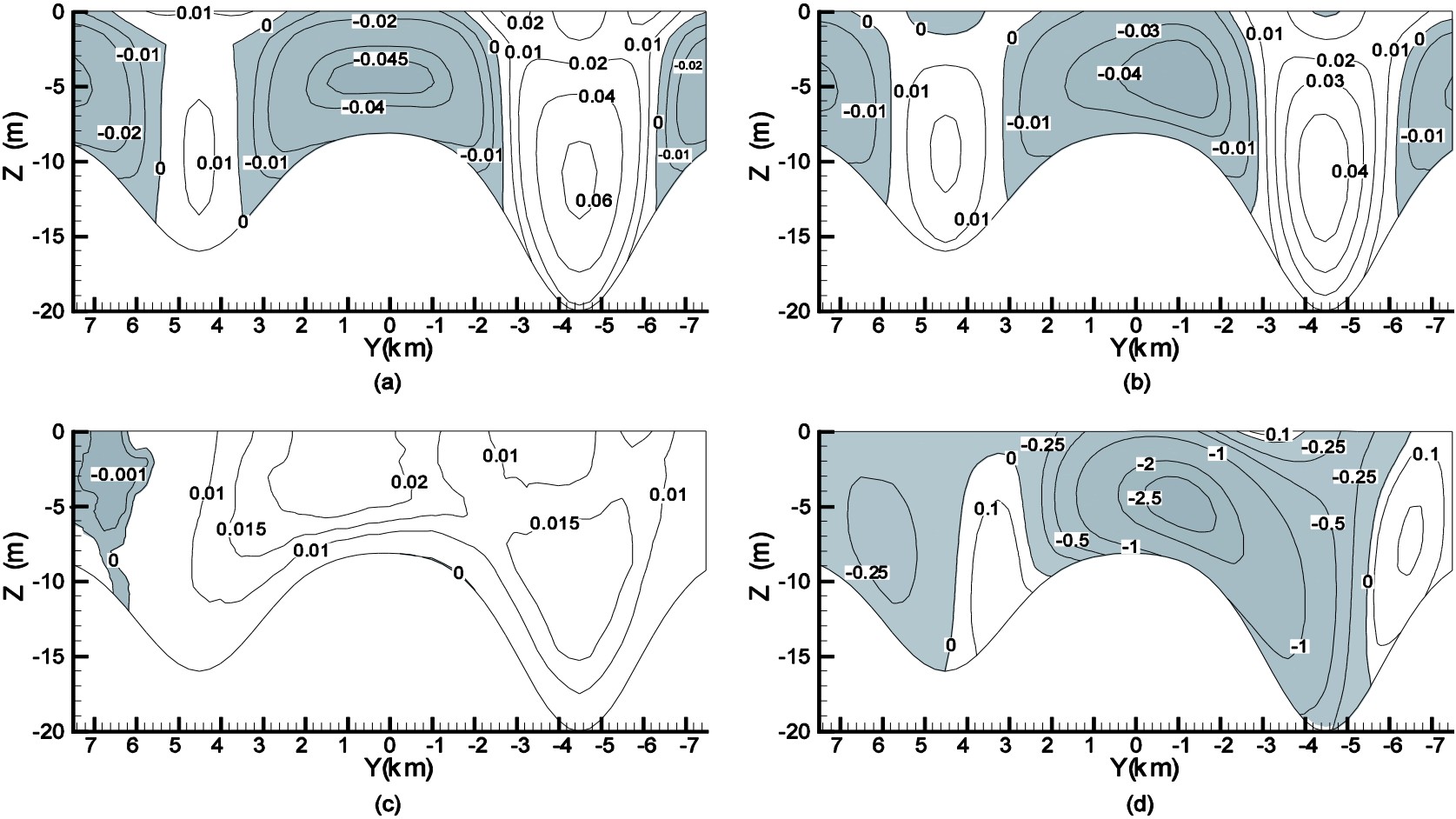

**Figure 7.** Contours of $u_x$ velocity (m/s) at (**a**) $x$ = 50 km and (**b**) $x$ = 98 km and contours of $u_y$ velocity (m/s $\times 10^{-2}$) at (**c**) x = 50 km and (**d**) $x$ = 98 km for the wind-density-driven case. The dark areas represent negative velocities.

The simulation time for each case was 47.3 h on a single core of the CPU and 1.68 h on the Tesla GPU, demonstrating a 28.16-fold speedup.

## 7. Conclusions

The lattice Boltzmann model (LBM) with multiple-relaxation-time (MRT) collision operation is a potential numerical method to simulate shallow water flow. The MRT-LBM can handle very low kinematic viscosity by using the last two relaxation rates (reciprocal of relaxation time) in D2Q9 lattice. Other relaxation rates can be determined with flexibility to ensure solution stability and accuracy. The MRT-LBM can avoid a stability problem which is often encountered when LBM with single-relaxation-time collision operation is used and the relaxation time is very close to 0.5.

The multilayer MRT-LBM is able to solve the multilayer Saint-Venant equations to obtain horizontal flow velocity distributions at different depths. The implementation of the multilayer MRT-LBM along with a given initial condition, boundary conditions, and forcing terms is straightforward. This study demonstrated the MRT-LBM to irregular bathymetry. The MRT-LBM solutions compare well to analytical solutions for horizontal velocity in various depths under the effects of wind-driven forcing, density-driven forcing, and their combined forcing.

The multilayer MRT-LBM is suitable for graphics processing unit (GPU) computing due to the locality of particle interaction and the transport of particle information in the LBM algorithm. For small grid sizes, the speedup is not impressive because the data transferring overhead between grid blocks on the GPU is not small compared to the actual computational cost. However, as the grid size increases, the computational cost becomes larger and dominates the data transferring overhead. This results in large speedup.

**Author Contributions:** Conceptualization, K.T. and F.T.; methodology, K.T. and F.T.; software, K.T.; validation, K.T. and F.T.; formal analysis, K.T. and F.T.; investigation, K.T. and F.T.; resources, F.T.; data curation, F.T.; writing—original draft preparation, K.T. and F.T.; supervision, F.T.; project administration, F.T.; funding acquisition, F.T.

**Funding:** The study was supported in part by the U.S. Geological Survey under Grant No. G16AP00056. The first author was also supported by the U.S. National Science Foundation under Grant No. DGE-0504507.

**Conflicts of Interest:** The authors declare no conflict of interest.

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
