# Peer review of "MRT-Lattice Boltzmann Model for Multilayer Shallow Water Flow"

_water, doi:10.3390/w11081623_

Round 1

Reviewer 1 Report

The manuscript deals with the application of the Multiple Relaxation Time Lattice Boltzmann (MRT LB) to the multilayer shallow water equations.

The manuscript is very interesting and well written.

My recommendation is to accept it for the publication on Water, after a very minor revision, which will further improve the quality of the manuscript, will have been performed.

The Authors should account for the difficulty of the Lattice Boltzmann in simulating critical and supercritical shallow water flows: i.e. when the Froude number is equal or greater than 1. Such an issue has been treated recently in La Rocca, M., Montessori, A., Prestininzi, P., & Succi, S. (2015). A multispeed Discrete Boltzmann Model for transcritical 2D shallow water flows. JOURNAL OF COMPUTATIONAL PHYSICS, 284, 117-132.

Moreover the citation of the application of the LB to environmental flows in estuarhine zones should be accounted for (Simulation of arrested salt wedges with a multi-layer Shallow Water Lattice Boltzmann model

P Prestininzi, A Montessori, M La Rocca, G Sciortino

Advances in water resources 96, 282-289)

As for citation  [34] the correct names are: La Rocca, M., C. Adduce, V. Lombardi, G. Sciortino, and R. Hinkelmann, 2012, Development of a lattice Boltzmann method for two-layered shallow-water flow: International Journal for Numerical Methods in Fluids, v. 70, no. 8, p. 1048–1072, doi:10.1002/fld.2742.

Please add the definition of matrix S.

Does the definition of the equilibrium PDF (19) coincide with the classical one of Zhou?

In the proposed case the Authors use the density value: 1025. Does it mean that salty water is considered? Why?

In the considered cases it is hard to appreciate the difference among the results obtained with different numbers of layers. In other words the difference among 5, 10 and 20 layers seems very small. The Authors should better comment this aspect.

Are there reference results for the last considered case (Wind-Driven and Density-Driven Circulation in Rotating Basins) or are the shown results  only numerical MRT LB?

Reviewer 2 Report

In this paper the authors introduce a multiple-relaxation-time lattice Boltzmann model (MRT-LBM) to simulate multilayer 2D shallow water flows using graphics processing unit computing to accelerate the lattice Boltzmann model. This approach allows them to handle low kinematic viscosity without causing a stability problem in the governing equations. Numerical solutions of the multi-layer MRT-LBM are verified by previous results for cases of wind-driven, density-driven, and combined circulations with non-uniform bathymetry. The results show good speedup and scalability for large problems.

I find the manuscript interesting and ground-breaking. The text is well written and the figures are clear. Although I believe the displayed results deserve publication, I would like the authors to clarify the following points:

1. The introduction is the enjoyable reading and let the reader to be introduced in the state of the art of the techniques employed in this paper. However, in my opinion it makes sense to provide some important references concerning advanced depth-averaged models for shear shallow water flows. For instance, in [A] and [B] the mathematical model of spatial shear shallow water flows of constant density is derived and studied. A two-layer long-wave approximation of the homogeneous Euler equations for a free-surface flow evolving over mild slopes is derived and verified in [C]. A new approach to model coastal waves in the shoaling and surf zones is presented in [D].

2. At the beginning of section 2, the authors state that equations (1)–(2) follow from the depth-averaging of the Navier-Stokes equations in the shallow water approximation (flow regime in which the vertical length scale is much smaller than the horizontal length scale). Due to the presence of viscosity, I believe that additional assumptions were used in the model derivation. For the same reason (viscous flow), the multilayer model (3)–(4) does not follow directly from equations (1)–(2). I would suggest giving a more accurate explanation of the model used or simply saying that viscous terms are added to regularize multilayer shallow water equations for inviscid flows.

3. Several external forces are included in the model. Some of them are fairly common (for example, bottom topography), but some require further explanation. In particular, the governing equations do not include the density ρ, so it should be constant. However, this variable and its derivatives appear in formula (10). In addition to the already defined kinematic viscosity ν, formula (8) includes another kinematic viscosity µ. What is the difference between them? Bottom friction is usually proportional to the square of the fluid velocity (like this -κv|v|), but here the linear law of friction is chosen.

4. Comparison of numerical simulation results for different numbers of layers with an analytical solution [48] is given in Section 6. Perhaps, this is a well-known solution for the lattice Boltzmann community, but for the other readers it would be useful to present the formulas of this solution and indicate the model for which these formulas are the analytical solution.

5.  Velocity profiles shown in figures 3 and 4 are not convex. As it is known, this can lead to a loss of stability of the shear shallow water flow with a free surface (see [E] for instance). Do the stability properties of the numerical algorithm decrease in such cases?

6. The results of the presented calculations look reasonable, but it would be nice to include comparison against laboratory data and/or field measurements. Is it possible to do this?

[A] V.M. Teshukov, Gas dynamic analogy for vortex free-boundary flows, J. Appl. Mech. Tech. Phys. 48 (2007) 303–309.

[B] S. Gavrilyuk, K. Ivanova, N. Favrie, Multi-dimensional shear shallow water flows: Problems and solutions, J. Comput. Phys. 366 (2018) 252–280.

[C] S.L. Gavrilyuk, V.Yu. Liapidevskii, A.A. Chesnokov, Spilling breakers in shallow water: applications to Favre waves and to the shoaling and breaking of solitary waves, J. Fluid Mech. 808 (2016) 441–468.

[D] M. Kazakova, G.L. Richard, A new model of shoaling and breaking waves: one-dimensional solitary wave on a mild sloping beach, J. Fluid Mech. 862 (2019) 552–591.

[E] A.A. Chesnokov, G.A. El, S.L. Gavrilyuk, M.V. Pavlov, Stability of shear shallow water flows with free surface, SIAM J. Appl. Math. 77 (2017) 1068–1087.

As a conclusion, I think that this paper should be accepted for publication after these enquiries are addressed in the revised paper.
